# UNIFIED IN-CONTEXT VIDEO EDITING

**Zixuan Ye**[1]* **Xuanhua He**[1]* **Quande Liu**[2]† **Qiulin Wang**[2] **Xintao Wang**[2]
**Pengfei Wan**[2] **Di Zhang**[2] **Kun Gai**[2] **Qifeng Chen**[1] **Wenhan Luo**[1]†
[1]The Hong Kong University of Science and Technology
[2]Kuaishou Technology
`https://zixuan-ye.github.io/UNIC`

## ABSTRACT

Recent advances in text-to-video generation have sparked interest in generative video editing tasks. Previous methods often rely on task-specific architectures (e.g., additional adapter modules) or dedicated customizations (e.g., DDIM inversion), which limit the integration of versatile editing conditions and the unification of various editing tasks. In this paper, we introduce UNified In-Context Video Editing (UNIC), a simple yet effective framework that unifies diverse video editing tasks within a single model in an in-context manner. To achieve this unification, we represent the inputs of various video editing tasks as three types of tokens: the source video tokens, the noisy video latent, and the multi-modal conditioning tokens that vary according to the specific editing task. Based on this formulation, our key insight is to integrate these three types into a single consecutive token sequence and jointly model them using the native attention operations of DiT, thereby eliminating the need for task-specific adapter designs. Nevertheless, direct task unification under this framework is challenging, leading to severe token collisions and task confusion due to the varying video lengths and diverse condition modalities across tasks. To address these, we introduce task-aware RoPE to facilitate consistent temporal positional encoding, and condition bias that enables the model to clearly differentiate different editing tasks. This allows our approach to adaptively perform different video editing tasks by referring the source video and varying condition tokens "in context", and support flexible task composition. To validate our method, we construct a unified video editing benchmark containing six representative video editing tasks. Results demonstrate that our unified approach achieves comparable performance with task specialists and exhibits emergent task composition abilities.

## 1 INTRODUCTION

Recent years have witnessed significant advancements in text-to-video foundation models based on diffusion (Chen et al., 2023; Wang et al., 2025a; Hong et al., 2022; Yang et al., 2024b; HaCohen et al., 2024; kli, 2025), establishing powerful tools for creating video content. Beyond the generation, video editing emerges as a natural extension, which aims to re-generate a reference video under multi-modal conditions, often incorporating fine-grained control signals like subject ID (He et al., 2024; Zhang et al., 2025; Yuan et al., 2024) and artistic style (Liu et al., 2023) along with textual prompts. Video editing spans diverse tasks, including global editing like style transfer (Liu et al., 2023; Ye et al., 2024), video propagation (Liu et al., 2024a; Ku et al., 2024), and local editing like object insertion, removal, swap (Tu et al., 2025; Zi et al., 2025; Lee et al., 2025), as well as video re-rendering like re-camera control (Bai et al., 2025a). These tasks hold vast potential for applications in film production, virtual reality, and automated content creation.

Current video editing methods primarily follow two strategies to inject reference video and control signals. As depicted in Fig. 2, one stream of methods, represented by Video-P2P (Liu et al., 2024b), AnyV2V (Ku et al., 2024), and FLATTEN (Cong et al., 2023), utilizes DDIM inversion for noise initialization to preserve the main structure of the reference video. However, these methods often fail to achieve ideal results and will inevitably introduce an additional stage, doubling the inference

---

*Equal contribution. Work done during an internship at KwaiVGI, Kuaishou Technology.
†Corresponding author.

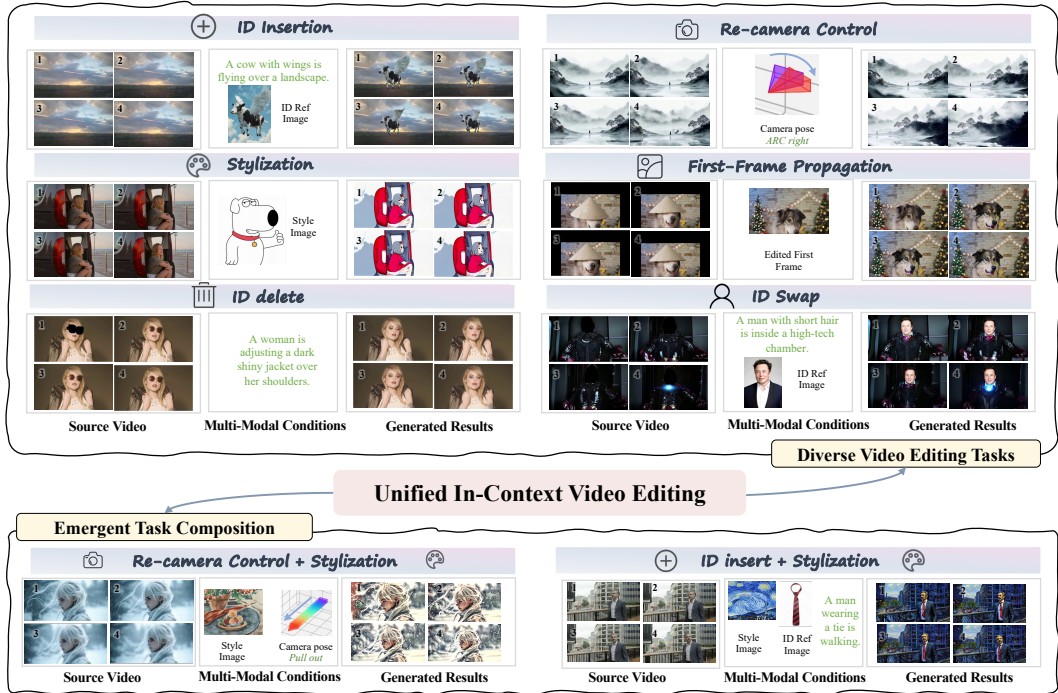

Figure 1: **Unified In-Context Video Editing enables unified video editing and emergent task composition.** Here we show the unification of six representative tasks, including ID Insert/Delete/Swap, Re-Camera Control, Stylization, and Propagation.

steps and cost. Another stream generally employs adapter-based designs (Ku et al., 2024; Jiang et al., 2025; Wang et al., 2024b; Zhang et al., 2023b; Mou et al., 2024) to inject different conditions, including reference video and multiple control signals. Despite promising progress, these methods suffer from two main challenges: 1) the adapter-based designs require modification to the model architectures and introduce parameter redundancy; and 2) these methods are generally task-specific, requiring training separate modules for each condition signal, raising difficulty for task extendability and unification. Very recently, VACE (Jiang et al., 2025) tries to categorize condition signals into frames and masks for unified video editing, yet still requires heavy adapter designs and is limited to process only visual conditions.

Based on these problems, this paper presents a unified and efficient framework for video editing tasks from multi-modal signals, named UNified In-Context Video Editing (UNIC). Inspired by the recent advancements in large language models and visual content generation (Yang et al., 2024a; Bai et al., 2024b; 2025b; Wang et al., 2025d; Chen et al., 2024c; Song et al., 2025; Tan et al., 2024; Xiao et al., 2024b), our key insight is to integrate diverse input signals from various editing tasks as a combined token sequence along the frame dimension, which are jointly modeled using the native transformer attentions to learn editing tasks from diverse context conditions. As shown in Fig. 2, to achieve unified video editing, UNIC formulates the inputs of different video editing tasks as three kinds of tokens, i.e., *1) the VAE tokens from reference video*, *2) the multi-modal condition signals* that vary upon the editing tasks, as well as *3) the noisy video latent*. By jointly concatenating these tokens and dynamically varying the condition token "in context", UNIC can flexibly perform diverse editing tasks without any architectural changes.

Crucially, directly concatenating these diverse input tokens presents unavoidable challenges for unified video editing. Firstly, the multi-modal conditions from different task types present inconsistent lengths, raising difficulty in achieving correct alignment with the video. For example, camera poses usually have a direct frame-to-frame correspondence to each video frame, while the style images directly affect the entire video. Such inconsistency makes it challenging to deal with varying-length video editing and leads to inevitable index collisions. Therefore, we propose Task-aware RoPE, which dynamically assigns unique Rotary Positional Embedding (RoPE) indices based on different task

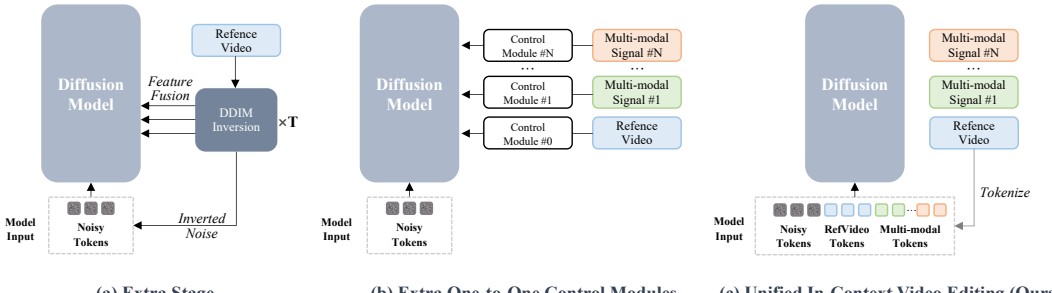

Figure 2: **Architectural comparison for incorporating conditioning signals. (a) Extra Stage:** Utilizes DDIM inversion on a reference video to derive inverted noise. **(b) Extra One-to-One Control Modules:** Employs dedicated, separate modules to process each control signal (e.g., reference video, multi-modal signals) and inject guidance into the diffusion model. **(c) In-Context Video Editing (Ours):** Our proposed method directly integrates guidance by tokenizing all conditioning signals (reference video, multi-modal signals) and concatenating them with the noisy input tokens, allowing the diffusion model to process all information jointly within its input sequence.

types, ensuring coherent temporal understanding regardless of varying condition length. Furthermore, different editing tasks may share the same modality of conditions (e.g., an image may represent an object identity in object editing or style in video stylization), leading to task confusion. To this end, we introduce a learnable condition bias for multi-modal condition signals, which enables the model to adaptively learn the target task type and resolve task ambiguity.

To validate the performance of our proposed framework, we construct a unified video editing benchmark incorporating six representative video editing tasks with distinct editing area ratio and conditioning modalities, including: local editing tasks of ID Swap/Delete/Insert (Jiang et al., 2025; Bian et al., 2025b); global editing tasks of stylization (Ye et al., 2024) and propagation (Liu et al., 2024a); and the re-rendering task of re-camera control (Bai et al., 2025a). These tasks exhibit significant variation in their input modalities (including text, images, and camera poses). Experimental results indicate that, despite the wide range of differences among these tasks, our framework not only successfully unifies them but also delivers superior performance across all. As shown in Fig. 1, our method offers two distinct advantages: 1) supporting a variety of editing tasks within a single framework without necessitating architectural changes, showcasing high flexibility; and 2) emergent capability to combine various editing tasks, highlighting its potential to unlock more complex and creative editing possibilities. We also provide in-depth analysis for unified video generation, respectively about the advantages of unified training over single task training as well as the training order of different tasks.

## 2 RELATED WORK

### 2.1 VIDEO EDITING AND RE-RENDERING

Based on current excellent T2V base models like Wan (Wang et al., 2025a), LongCat(Team et al., 2025), the editing capability of these models is being actively explored and has attracted significant attention. Video editing encompasses diverse tasks, ranging from local adjustments to global transformations. Achieving these modifications often requires injecting multi-modal signals, such as motion (Tu et al., 2025; Wang et al., 2024c; 2025b; Xiao et al., 2024a; Wei et al., 2024), style (Liu et al., 2023; Ye et al., 2024; Yang et al., 2025b), object attributes (Kong et al., 2024; Huang et al., 2025; He et al., 2024; Zhang et al., 2025), audio (Kong et al., 2025b; Yang et al., 2025a), or camera pose (Bai et al., 2025a; 2024a; Wang et al., 2024c; Luo et al., 2025), into generation process.

To preserve information from the reference video during editing, several methods employ DDIM inversion. This technique initializes generation noise based on the reference video and injects features extracted during inversion into the denoising steps. For instance, VideoP2P (Liu et al., 2024b) copies inversion features and replaces specific cross-attention maps to align with editing requirements. FLATTEN (Cong et al., 2023) utilizes optical flow to identify keypoints and inject their features to

maintain motion fidelity. AnyV2V (Ku et al., 2024) leverages spatial, temporal, and CNN features gathered during inversion. While these approaches excel at retaining reference video information, they require an **additional inversion stage**, increasing inference cost and computational overhead. To reduce computational cost, image editing works like FlowEdit (Kulikov et al., 2025) and InfEdit (Xu et al., 2024) explore inversion-free methods. Similarly, video editing methods like FlowDirector (Cai et al., 2025; Zhang & Han; Li et al., 2025b) follow this insight and apply it to video editing. However, they can only perform limited video editing scenarios like text-based attribute editing, while IF-V2V (Kong et al., 2025a) can only perform I2V propagation, which is not sufficient to cover and address various video editing tasks.

Instead of additional processing stages, another strategy focuses on injecting control directly into the denoising network using auxiliary modules. Some works adopt an extra model to process diverse conditions, for example VEGGIE (Yu et al., 2025) uses a grounding model for processing, while UniVideo (Wei et al., 2025) adopts an LLM as text encoder to enhance the generation performance. Another stream of work assigns a control module for each condition. For example, to maintain structural or layout information of reference videos, Follow-your-Canvas (Chen et al., 2024a) extracts window details with a layout encoder, while MagicEdit (Liew et al., 2023) leverages the depth video of the reference video via a depth ControlNet (Zhang et al., 2023a). For finer-grained preservation of content and motion details, VideoAnyDoor (Tu et al., 2025) and Revideo (Mou et al., 2024) use separate encoders to obtain the feature and inject them through ControlNet. To incorporate additional multi-modal conditions, VideoAnyDoor employs an extra ID encoder, infusing identity information via cross-attention. Likewise, StyleMaster (Ye et al., 2024) introduces a dedicated style encoder, injecting style features through cross-attention. A common choice of these methods is the reliance on specialized control modules for each condition type. This design choice, requiring **additional control modules for different conditions**, inevitably increases overall model complexity and limits the extensibility to diverse or novel video editing tasks.

## 2.2 Universal Generative Models

Developing unified "omni" solutions capable of handling diverse generative tasks within a single model is a challenging but highly valuable goal. The field of image editing and generation has witnessed a clear trend towards such unification. Early works like Instruct-imagen (Hu et al., 2024) integrated multi-modal instructions using cross-attention. OmniGen (Xiao et al., 2024b) further advances this by tokenizing all conditions as direct inputs to the transformer, creating a flexible "any-purpose" generation model without external plugins. This powerful approach of handling conditions directly within the core architecture has been validated and extended by subsequent research, including OminiControl (Tan et al., 2024), ACE (Han et al., 2024), UniReal (Chen et al., 2024b), and other unified models (Le et al., 2024; Song et al., 2025; Mou et al., 2025).

In contrast, unified video generation approaches often depend on task-specific control mechanisms. VideoComposer (Wang et al., 2024b) employs different ControlNets for various inputs, and VACE (Jiang et al., 2025) uses specialized control blocks. Drawing inspiration from image editing paradigms that leverage full-attention mechanisms to replace dedicated modules like ControlNet, we propose extending this strategy to video editing. Although FullDiT (Ju et al., 2025) demonstrated 3D full-attention potential for multi-control video generation, the leap to more difficult video editing remains an open challenge. Our work aims to fill this gap by **developing a more unified and flexible method for general video editing purposes**, eliminating the need for separate control modules.

## 3 Method

Towards general video editing, we first review the diverse video editing tasks, and systematically define all inputs across different tasks into three basic types. Building on this, we introduce an in-context video editing framework, offering a parameter-efficient and highly flexible approach adaptable to various editing purposes. We further elaborate on the specific design within our architecture for task differentiation and flexibility.

### 3.1 Multi-modal Driven Video Editing Tasks

Multi-modal conditions driven video editing tasks, as illustrated in Fig. 3, incorporating various inputs. For example, stylization requires a reference video and a style image, while object insertion needs a reference video, text, and an object image. Fundamentally, these tasks are driven by unique combinations of multi-modal inputs. We generalize these inputs into three basic types: **1) Noisy tokens** represent the initial latent state of the target video, typically from random noise or a noise-added input video latent. **2) Reference video tokens** represent the VAE tokens of a reference video, providing crucial temporal context, motion, and visual content. Their influence varies with the task's alignment requirements. For strict-alignment tasks like ID deletion or stylization, these tokens enforce strong frame-by-frame correspondence, ensuring the output precisely follows the motion and unedited content. In soft-reference scenarios such as re-camera control, these tokens provide more abstract guidance, following the overall content or motion style without demanding strict pixel-level matching, which allows for significant deviations. **3) Multi-modal condition tokens** include all other forms of guidance signals. This versatile category includes image tokens and auxiliary control tokens. For example, image tokens encode reference images, guiding structural edits (like the edited first frame in video propagation) or serving as a style reference (in stylization) and ID reference (in ID insertion). Besides, auxiliary control tokens can contain diverse signals like camera poses, depth maps, segmentation masks, human pose skeletons, edge maps, sparse trajectories for motion guidance, and audio signals for lip-syncing, providing fine-grained control.

This classification allows any video editing task to be represented within our structured approach, the specific combination can leads to different tasks, making it easier to uniformly process these tasks within a unified framework.

### 3.2 Unified In-Context Video Editing

We introduce Unified In-Context Video Editing (UNIC), a simple yet effective paradigm for diverse video editing tasks. The core idea is to represent all inputs, including the noisy tokens, the reference video, and all other multi-modal conditioning signals (images, cameras, etc.), as a single, unified token sequence. This contrasts with approaches that inject conditions via additional control modules. By processing all information jointly within the full-attention layers, the model can flexibly learn the **contextual relationships** for different editing tasks from the conditions provided "in context".

**Preliminary.** Our method inherits the video diffusion transformers trained using flow matching. Specifically, the training objective is given by:

$$\mathcal{L}_{\text{FM}}(\theta) = \mathbb{E}_{t,\boldsymbol{x}_0,\boldsymbol{x}_1} \left\| \boldsymbol{v}_\theta(\boldsymbol{x}_t, t) - (\boldsymbol{x}_1 - \boldsymbol{x}_0) \right\|_2^2, \tag{1}$$

where $\boldsymbol{x}_1 \sim p(\boldsymbol{x}_1)$ represents the video sample, $\boldsymbol{x}_0 \sim \mathcal{N}(\boldsymbol{0}, \boldsymbol{1})$ is a Gaussian sample, and $t$ is randomly distributed in $[0, 1]$. The function $\boldsymbol{v}_\theta$ is the neural network that takes the noised version $\boldsymbol{x}_t = t\boldsymbol{x}_1 + (1 - t)\boldsymbol{x}_0$ as input.

Training with this loss function leads to the following ordinary differential equation (ODE):

$$\frac{d\boldsymbol{x}_t}{dt} = \boldsymbol{v}_\theta(\boldsymbol{x}_t, t), \tag{2}$$

which allows us to sample a synthesized video $\boldsymbol{x}_1$ from a random Gaussian noise $\boldsymbol{x}_0$.

Our transformer consists of several DiT blocks, where each block contains 2D self-attention to learn spatial information and 3D self-attention to fuse spatio-temporal information.

#### 3.2.1 In-Context Video Editing

For video editing and re-rendering tasks, given a reference video $V_{ref} \in \mathbb{R}^{f \times c \times h \times w}$ and a set of conditions $\{C_i \mid i = 1 \ldots n\}$, our goal is to generate a target video $V_{tar} \in \mathbb{R}^{f \times c \times h \times w}$ that aligns with these conditions while preserving required content of $V_{ref}$.

Instead of using DDIM inversion or adding add additional control modules like previous work, we propose a unified and parameter-efficient approach, concatenating all the inputs along the frame dimension to perform self-attention. First, we encode the reference video $V_{ref}$ using a 3D VAE

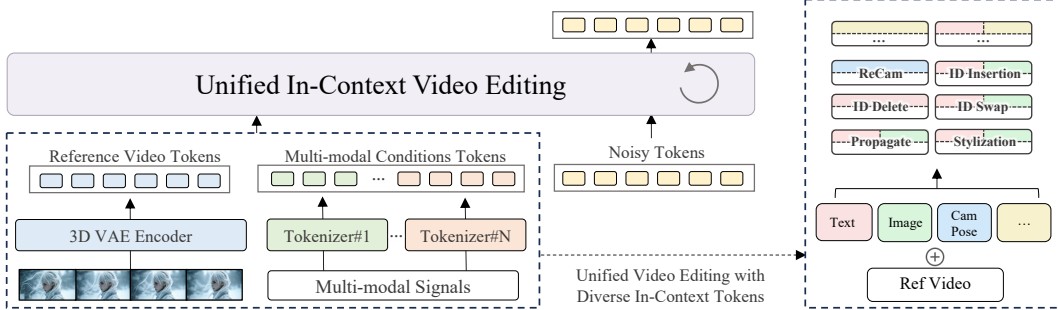

Figure 3: **Overall Pipeline of Unified In-Context Video Editing.** Our framework utilizes a unified transformer architecture for video editing. The model input is created by concatenating noisy tokens, reference video tokens, and multi-modal condition tokens (task-specific controls like images), these combined tokens form a single input sequence along the frame dimension. By simply modifying the multi-modal condition tokens, this framework can handle any video editing task.

encoder to obtain its latent representation $z_{ref}$. Similarly, other conditions $C_i$ are converted into token sequences $z_i$ using modality-specific tokenizers (e.g., the same 3D VAE for image conditions, a T5 tokenizer for text, an MLP for camera pose, etc.).

As shown in Fig. 3, during generation, the model operates on a noisy latent $z_{tar}$. By concatenating multi-modal tokens as: $z_{cond} = [z_1; \ldots; z_N]$, which is then combined with noisy token $z_{tar}$, reference video tokens $z_{ref}$ in frame dimension into a single sequence as the model input $z = [z_{tar}; z_{ref}; z_{cond}]$, we can simply perform full 3D attention to enable the interaction of the tokens. Without requiring an intricate inversion process or task-specific architectural modifications, the only requirements for this framework are the modality-specific tokenizers, which are also needed in other methods.

### 3.2.2 EFFECTIVE AND FLEXIBLE TASK UNIFICATION

Based on in-context editing framework, theoretically we can unify all the video editing tasks. However, directly concatenating them introduces specific challenges: **1) Task Ambiguity:** When different editing tasks rely on conditions from the same modality (an image can refer to style reference or ID), simple concatenation can make it difficult for the model to distinguish the target task of the conditional tokens. **2) Positional Encoding Conflicts & Inflexibility:** The 3D RoPE used in video generation base models, often assigns sequential indices in the frame dimension, which will raise problems when handling different conditions. For example, style reference does not have a direct correspondence with video frames, whereas camera pose requires frame-to-frame correspondence, making it difficult to maintain appropriate alignments under sequential indices. The situation becomes more problematic in variable-length editing, where it even struggles to clearly distinguish boundaries between reference videos and conditions. To address these difficulties, we introduce two key components: Condition Bias and Task-Aware RoPE Index.

**Condition Bias** As mentioned above, to address the task ambiguity, we propose *Condition Bias*, the task-specific learnable embedding that is directly added to tokens before attention computation. To be specific, for each condition in the context, including the multi-modal signals and reference video, i.e., $z_i \in \{z_{ref}, z_1, \ldots, z_N\}$, we inject a learnable bias $b_i \in \mathbb{R}^d$ corresponding to its task type:

$$\tilde{z}_i = z_i + b_i . \tag{3}$$

These task-aware tokens $\{\tilde{z}_i\}$ then undergo standard full self-attention. The biases implicitly guide attention by structuring token representations: tokens from the same task share similar bias-induced feature offsets, promoting intra-task alignment while maintaining cross-task distinction. The learnable embeddings are zero-initialized to preserve original token semantics, and their dimension-preserving addition enables simple integration with existing architectures.

**Task-aware RoPE Index** Standard 3D Rotary Position Embedding (RoPE) assigns sequential indices to frames. For instance, in a re-camera control task with $N$ frames, noisy tokens might occupy indices 0 to $N-1$, reference video tokens $N$ to $2N-1$, and camera poses $2N$ to $3N-1$. When the

video frame length $N$ varies, it will be hard to find the boundaries between the conditions, causing poor frame alignment.

To address this, we introduce a task-aware RoPE indexing scheme: 1) For tasks where conditional inputs have a direct frame-to-frame correspondence with the video (e.g., reference videos, camera poses, audio), we reuse the indices of the noisy latent video frames (0 to $N - 1$), which can help maintain the alignment. 2) For tasks that do not have such direct frame-to-frame correspondence (e.g., ID images, style references), we assign indices based on a base offset and task-specific offset. The base offset, $m$, is determined by the video length (i.e., $m = N$). Beyond this, each task $t$ is assigned with a pre-defined, fixed task offset, $O_t$, and a slot count (or length), $L_t$. The value $O_t$ dictates the starting point of task $t$'s allocation relative to $m$, and $L_t$ defines how many consecutive indices the task occupies. The index range for such a task $t$ is calculated as:

$$\text{Index}(t) = (m + O_t) + [0, \dots, L_t - 1] \qquad (4)$$

These task offsets $O_t$ and lengths $L_t$ are chosen to ensure non-overlapping task slots and maintain clear task distinction. For example, if ID images (task 1) are defined with a task offset $O_1 = 100$ and require $L_1 = 3$ slots (supporting up to 3 IDs), they will use indices from $N + 100$ to $N + 102$. If a style reference (task 2) is defined with a task offset $O_2 = 200$ and requires $L_2 = 1$ slot, it will occupy the index $N + 200$. This adaptive approach ensures that task slots automatically scale with the video length $N$, while preserving their relative positional relationships and avoiding overlap.

**Training Strategy**  In this work, we unify six tasks into this framework, including ID insert, ID swap, ID delete, re-camera control, video style transfer, video propagation. These tasks were chosen due to their diverse condition modalities and varying editing degree. However, joint training performance is affected by training order due to differences in task complexity and convergence rates. For example, re-camera control requires 600k iterations to converge, while simpler tasks like ID-swapping achieve good results in only 80k iterations. During joint training, if we start with easier tasks, they dominate the optimization with faster loss reduction, failing to allocate sufficient capacity to more difficult tasks. Therefore, we choose a hard-to-easy training strategy, i.e., progressively training tasks from higher to lower difficulty for convergence.

## 4 EXPERIMENTS

**Dataset and Benchmark**  UNIC is trained on multiple datasets to support multi-task video editing. For ID swap/insert/delete and stylization tasks, we use self-constructed datasets (see Appendix for details). These are also adapted for the propagation task by using the first frame of the target video as the edited input. For re-camera control, we use the Multi-Cam Video Dataset from ReCamMaster (Bai et al., 2025a). To comprehensively evaluate the method, we construct **a unified video editing benchmark** incorporating six representative video editing tasks with distinct editing area ratio and conditioning modalities. The details of the benchmark can be found in Appendix.

**Evaluation Metrics**  The evaluation of these tasks is conducted across two dimensions: task-specific performance and overall video quality. To assess task-specific performance, for ID tasks, we employ the DINO-score (Caron et al., 2021) and CLIP-score (Radford et al., 2021) to evaluate identity similarity with the reference image. For the style task, style similarity with the reference style image is validated using the CSD-score (Somepalli et al., 2024), while ArtFID and CFSD (Chung et al., 2024) are used to consider content preservation. Furthermore, for re-camera control, we use RotErr, TransErr, and CamMC following CamI2V (Zheng et al., 2024) to evaluate the alignment with the given camera pose. Besides, the overall quality of the generated videos is assessed by motion smoothness, dynamic degree, and aesthetic score, subject consistency, background consistency, temporal flickering and imaging quality (Huang et al., 2024).

### 4.1 EDITING PERFORMANCE COMPARISON

We compare our approach with the state-of-the-art unified video editing methods like VACE (Jiang et al., 2025) and task-specific methods from different video editing methods, like ReCamMaster (Bai et al., 2025a). We provide an evaluation of overall video performance, including text alignment and overall quality, as well as task-specific metrics such as DINO-score in ID-related tasks. Since

Table 1: Quantitative comparison on six video editing tasks: ID Insert/Swap/Delete, Re-Camera control, Stylization, and Propagation. Best results are highlighted in **bold**.

**ID Insert**

| Method | Identity | | Alignment | Video Quality | | |
|---|---|---|---|---|---|---|
| | CLIP-I↑ | DINO-I↑ | CLIP-score↑ | Smoothness↑ | Dynamic↑ | Aesthetic↑ |
| VACE (Jiang et al., 2025) | 0.522 | 0.110 | 0.100 | 0.933 | **44.568** | 5.407 |
| Pika (pik, 2025) | **0.689** | **0.387** | **0.253** | 0.934 | 20.65 | 5.393 |
| Ours | 0.598 | 0.245 | 0.216 | **0.961** | 11.07 | **5.627** |

**ID Swap**

| Method | Identity | | Alignment | Video Quality | | |
|---|---|---|---|---|---|---|
| | CLIP-I↑ | DINO-I↑ | CLIP-score↑ | Smoothness↑ | Dynamic↑ | Aesthetic↑ |
| VACE (Jiang et al., 2025) | 0.712 | 0.423 | 0.230 | 0.964 | **29.306** | 6.015 |
| Pika (pik, 2025) | 0.700 | 0.393 | 0.209 | 0.948 | 20.09 | 5.210 |
| AnyV2V(Prop) (Ku et al., 2024) | 0.605 | 0.229 | 0.218 | 0.917 | 7.596 | 4.842 |
| Ours(Prop) | 0.693 | 0.414 | 0.236 | **0.980** | 5.153 | 5.801 |
| Ours | **0.725** | **0.429** | **0.242** | 0.971 | 7.500 | **6.056** |

**ID Delete**

| Method | Video Reconstruction | | Alignment | Video Quality | | |
|---|---|---|---|---|---|---|
| | PSNR↑ | RefVideo-CLIP↑ | CLIP-score↑ | Smoothness↑ | Dynamic↑ | Aesthetic↑ |
| AnyV2V(Prop) (Ku et al., 2024) | 19.504 | 0.869 | 0.205 | 0.964 | 4.980 | 5.325 |
| VACE (Jiang et al., 2025) | 20.947 | 0.883 | 0.211 | 0.966 | **15.441** | 5.332 |
| VideoPainter (Bian et al., 2025b) | **22.987** | **0.920** | 0.212 | 0.957 | 13.759 | 5.403 |
| Ours(Prop) | 20.378 | 0.906 | 0.209 | 0.968 | 9.017 | 5.408 |
| Ours | 19.171 | 0.900 | **0.217** | **0.970** | 10.934 | **5.493** |

**Propagation**

| Method | Frame Alignment | Alignment | Video Quality | | |
|---|---|---|---|---|---|
| | RefVideo-CLIP↑ | CLIP-score↑ | Smoothness↑ | Dynamic↑ | Aesthetic↑ |
| AnyV2V (Ku et al., 2024) | 0.812 | 0.229 | 0.935 | 13.462 | 5.136 |
| VACE(I2V) (Jiang et al., 2025) | 0.574 | **0.234** | 0.932 | **36.783** | 5.425 |
| Ours | **0.840** | 0.226 | **0.966** | 12.762 | **5.565** |

**Stylization**

| Method | Style & Content | | Alignment | Video Quality | | |
|---|---|---|---|---|---|---|
| | CSD-Score↑ | ArtFID↓ | CLIP-score↑ | Smoothness↑ | Dynamic↑ | Aesthetic↑ |
| AnyV2V(Prop) (Ku et al., 2024) | 0.207 | 43.299 | 0.195 | 0.937 | 9.227 | 4.640 |
| StyleMaster (Ye et al., 2024) | **0.306** | 38.213 | 0.188 | **0.952** | 9.758 | 5.121 |
| Ours(Prop) | 0.197 | **36.198** | **0.215** | 0.932 | **11.569** | 5.045 |
| Ours | 0.259 | 37.619 | 0.171 | 0.945 | 9.370 | **5.276** |

**Re-Camera Control**

| Method | Camera Control | | Alignment | Video Quality | | |
|---|---|---|---|---|---|---|
| | RotErr↓ | TransErr↓ | CLIP-score↑ | Smoothness↑ | Dynamic↑ | Aesthetic↑ |
| ReCamMaster-Wan (Bai et al., 2025a) | 1.454 | 5.695 | 0.219 | 0.917 | **31.751** | 4.738 |
| Ours | **1.275** | **5.667** | **0.220** | **0.933** | 24.21 | **4.826** |

the propagation version (indicated by a (prop) suffix) for Swap/Delete/Stylization tasks requires the edited first frame, we specifically use Insert-Anything (Song et al., 2025), FLUX (Labs, 2024), and the first frame of StyleMaster (Ye et al., 2024) to obtain the edited first frame.

As presented in Table 1, our six-in-one framework demonstrates consistent and strong performance across all evaluated video editing tasks. Notably, our model achieves leading results in ID Insert and Re-Camera Control, outperforming existing methods on most metrics. For Stylization, we achieve comparable performance to specialized models like StyleMaster (Ye et al., 2024). While for ID Delete, specialized models like VideoPainter (Bian et al., 2025b) show better video reconstruction with PSNR, our approach still surpasses it in alignment with CLIP-score and some video quality aspects such as Smoothness and Aesthetic score. Furthermore, a significant advantage of our method is its flexibility, supporting arbitrary resolutions and lengths, a capability not present in many fixed-length video editing models.

## 4.2 COMPARISON WITH OTHER CONDITION INJECTION METHODS

To further demonstrate the difference with other conditioning and tuning methods, we implement both LoRA and adapter-based versions, with the latter following the design of VideoAnyDoor (Tu et al., 2025) for ID insert and ID swap tasks. The computational cost and performance are shown in Table 2 and Table 3. Our method demonstrates superior performance compared to other approaches, consistently outperforming both LoRA and Adapter-based methods across all metrics in both ID Insert and ID Swap tasks. Moreover, with a simple step cache optimization, our approach shows significant efficiency potential while maintaining competitive performance.

Table 2: Computational cost comparison with LoRA and adapter-based methods on ID Insert task.

| Method | FLOPs | Params | CLIP-I↑ | DINO-I↑ | CLIP-Score↑ |
|---|---|---|---|---|---|
| Adapter-based | 51.73T | 1.65B | 0.505 | 0.190 | 0.158 |
| LoRA | 69.29T | 1.20B | 0.537 | 0.222 | 0.193 |
| Full Finetuning | 69.29T | 1.20B | 0.528 | 0.242 | 0.203 |
| Ours | 69.29T | 1.20B | 0.568 | 0.254 | 0.227 |
| Ours + step cache | 39.28T | 1.20B | 0.558 | 0.249 | 0.232 |

Table 3: Quantitative comparison with LoRA and adapter methods.

| ID Insert | CLIP-I↑ | DINO-I↑ | CLIP-Score↑ |
|---|---|---|---|
| Adapter-based | 0.505 | 0.190 | 0.158 |
| LoRA | 0.537 | 0.222 | 0.193 |
| Ours | 0.568 | 0.254 | 0.227 |
| **ID Swap** | **CLIP-I↑** | **DINO-I↑** | **CLIP-Score↑** |
| Adapter-based | 0.694 | 0.417 | 0.221 |
| LoRA | 0.713 | 0.433 | 0.230 |
| Ours | 0.732 | 0.449 | 0.238 |

Table 4: **Ablation study on the training order of different tasks.** We employ three settings: hard to easy, easy to hard, and joint training. Best results are highlighted in **bold**.

| Training Order | Identities | | ReCam | | | Style | | |
|---|---|---|---|---|---|---|---|---|
| | CLIP-I↑ | DINO-I↑ | RotErr↓ | TransErr↓ | CamMC↓ | CSD-Score↑ | ArtFID↓ | CFSD↓ |
| -> camera
-> camera+id
-> camera+id+style+propagation | 0.725 | **0.429** | **1.275** | **5.667** | **6.154** | 0.259 | **37.619** | **0.107** |
| -> id
-> id+style+propagation
-> camera+id+style+propagation | **0.726** | 0.427 | 1.398 | 5.681 | 6.275 | 0.247 | 37.748 | 0.109 |
| -> camera+id+style+propagation | 0.713 | 0.421 | 2.287 | 9.694 | 10.377 | **0.298** | 38.953 | 0.170 |

## 4.3 ANALYSIS FOR OUR IN-CONTEXT VIDEO EDITING

We provide a set of experiments to demonstrate the training choice of in-context video editing for different tasks. Based on our experimental results, we summarize several key findings that highlight how specific training strategies enable robust multi-task unification and enhance the overall performance of our in-context framework.

**Should we train tasks sequentially or jointly?** To determine the optimal training strategy for our unified tasks, which possess varying levels of difficulty, we investigate whether sequential training or joint training yields superior performance. As shown in Table 4, we report results under different training strategies. Among our selected six tasks, re-camera control is the most difficult, since the modality is far away from the visual content, while the ID-related task is relatively easy to learn. We experiment with three approaches: (1) sequential training from hard to easy tasks, (2) sequential training from easy to hard tasks, and (3) joint training of all tasks from scratch. Our findings indicate that the sequential training approaches (hard-to-easy and easy-to-hard) can help the multi-task learning. In contrast, joint training from scratch, while capable of learning easier tasks, struggles significantly with harder ones, resulting in poor performance on the re-camera control task.

**Will the task unification affect single-task performance?** There is a natural concern about whether multi-task learning affects single-task performance. To explore this, we conduct a comparative study between task-specific models and the unified model. We train separate models for stylization, ID-related editing, and re-camera control, and compare their performance with the unified model. As shown in Table 5, the unified model does not impair task performance and even offers advantages in camera control and style similarity in stylization. However, it slightly compromises content preservation in stylization. This trade-off is due to the training mechanism: during ID-related tasks, the model is trained to fully preserve content. As a result, in stylization tasks, the unified model retains more information than just style, leading to higher style similarity but reduced content preservation. Overall, unifying diverse tasks within this framework does not significantly degrade individual task performance and can even enhance it in some cases.

**Do condition bias and task-aware RoPE Matter?** To validate our proposed Condition Bias and Task-Aware RoPE, we conduct an ablation study, with results presented in Table 6. Comparing our full model against variants clearly demonstrates their benefits. The baseline model (D1), lacking both components, performs adequately on simple tasks but struggles with complex temporal tasks like re-camera control, showing high TransErr and CamMC. Adding Condition Bias alone (D2) improves CLIP-I in ID swap and reduces TransErr in re-camera control, while implementing only Task-aware

Table 5: **Performance Comparison between task-specific model and unified model.** We evaluate our unified model (B4) against task-specific models trained for ID (B1), style (B2), and re-camera control (B3) on relevant metrics for each task. Best results are highlighted in **bold**.

| | Task | Identities | | Style | | | ReCam | | |
|---|---|---|---|---|---|---|---|---|---|
| | | DINO-I↑ | CLIP-I↑ | CSD-Score↑ | ArtFID↓ | CFSD↓ | RotErr↓ | TransErr↓ | CamMC↓ |
| B1 | id | **0.449** | 0.723 | - | - | - | - | - | - |
| B2 | style | - | - | 0.234 | 37.674 | **0.096** | - | - | - |
| B3 | camera | - | - | - | - | - | 1.472 | 5.836 | 6.434 |
| B4 | id+style+camera | 0.429 | **0.725** | **0.259** | **37.619** | 0.107 | **1.275** | **5.667** | **6.154** |

Table 6: **Ablation Study on Condition Bias and Task-aware RoPE.** We compare our full model (D4) with their variants. Best results are highlighted in **bold**. ↑ indicates higher is better; ↓ indicates lower is better.

| | Condition Bias | RoPE | | Identities | | Style | | | ReCam | | |
|---|---|---|---|---|---|---|---|---|---|---|---|
| | | Sequential | Task-aware | DINO-I↑ | CLIP-I↑ | CSD-Score↑ | ArtFID↓ | CFSD↓ | RotErr↓ | TransErr↓ | CamMC↓ |
| D1 | | ✓ | | 0.433 | 0.710 | 0.242 | 34.194 | 0.081 | 2.501 | 8.972 | 13.119 |
| D2 | ✓ | ✓ | | **0.434** | 0.723 | **0.274** | 35.548 | 0.091 | 1.428 | 6.039 | 6.566 |
| D3 | | | ✓ | 0.422 | 0.710 | 0.258 | **32.768** | **0.072** | 1.304 | 6.038 | 6.498 |
| D4 | ✓ | | ✓ | 0.429 | **0.725** | 0.259 | 37.619 | 0.107 | **1.275** | **5.667** | **6.154** |

RoPE (D3) significantly reduces CamMC in re-camera control. The full model (D4), combining both components, achieves superior performance across all tasks. These results demonstrate how Condition Bias enables effective task disambiguation while Task-aware RoPE enhances temporal modeling, together creating a robust unified video editing framework.

## 5 CONCLUSION

In this paper, we introduce UNified In-Context Video Editing (UNIC), a simple yet effective framework that unifies diverse video editing tasks within a single model in an in-context manner. To this end, we formulate the input of different video editing tasks as three types of tokens, integrating them as a single unified token sequence jointly modeled with the original full-attention of diffusion transformers. With the devised task-aware RoPE and conditional bias, our method can flexibly perform different editing tasks and support their combination. To facilitate the evaluation, we also construct a unified video editing benchmark. Extensive experiments on six representative video editing tasks demonstrate that our unified model shows superior performance on each task and exhibits emergent task composition abilities.

## 6 LIMITATION AND FUTURE WORK

Our current unification efforts are limited to the six tasks discussed. The incorporation of additional modalities, such as lip-syncing with audio (Kong et al., 2025b; Zhong et al., 2025; Zhang et al., 2024), visual effects (Li et al., 2025a; Bian et al., 2025a), and multi-shot scenarios (Wang et al., 2025c) remains unexplored in this work. In future research, we intend to integrate a wider array of tasks to investigate whether there is an upper limit to the number of tasks that can be successfully unified. Furthermore, recognizing that a high token count during self-attention can significantly increase computational overhead, we plan to explore architectural designs or alternative mechanisms to improve efficiency (He et al., 2025). Also, reinforcement learning can be used to enhance the generation quality and control accuracy (Liu et al., 2025; 2026; Ye et al., 2025; Qin et al., 2025).

## 7 ACKNOWLEDGMENT

This work was supported in part by the National Natural Science Foundation of China (Grant No. 62372480), and in part by the 2025 Tencent AI Lab Rhino-Bird Focused Research Program.

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

## A    APPENDIX

The Appendix of this paper contains the following main sections:

- Additional Experiment and Analysis - providing supplementary experimental results and detailed analysis
- UNIC Benchmark - describing the benchmark details
- Training Dataset Construction - the construction of training datasets
- Training Schemes - explaining the specific training strategies and protocols
- Limitation and Future Work - discussing current limitations and future research directions
- Statement for Large Language Models - clarifying the usage of large language models

## B    ADDITIONAL EXPERIMENT AND ANALYSIS

### B.1    VIDEO QUALITY COMPARISON ON DIFFERENT EDITING TASKS

Here we give a more comprehensive report on the generated video quality compared with the baseline methods, as shown in Table 7. Our method demonstrates consistently stable video quality across all editing tasks. The results show that our approach maintains reliable performance metrics across different scenarios, including identity swap, style transfer, and camera control tasks. The temporal quality remains stable with competitive motion smoothness and minimal flickering artifacts. Additionally, our method exhibits robust consistency in both subject and background preservation while delivering satisfactory imaging quality.

Table 7: Video Quality Comparison on different editing tasks. Best results are highlighted in **bold**.

| | Motion Smoothness | Dynamic Degree | Aesthetic Score | Subject Consistency | Background Consistency | Temporal Flickering | Imaging Quality |
|---|---|---|---|---|---|---|---|
| **ID Swap** | | | | | | | |
| VACE (Jiang et al., 2025) | 0.990 | **0.350** | **0.613** | 0.966 | 0.957 | 0.982 | 0.697 |
| Pika (pik, 2025) | **0.994** | 0.250 | 0.517 | 0.973 | 0.960 | **0.990** | **0.708** |
| AnyV2V(Prop) (Ku et al., 2024) | 0.966 | 0.150 | 0.499 | 0.873 | 0.909 | 0.957 | 0.552 |
| Ours | 0.993 | 0.100 | 0.605 | **0.975** | **0.970** | 0.988 | 0.690 |
| | | | | | | | |
| AnyV2V(Prop) (Ku et al., 2024) | 0.960 | 0.350 | 0.433 | 0.858 | 0.924 | 0.949 | 0.375 |
| StyleMaster (Ye et al., 2024) | 0.988 | **0.400** | 0.490 | **0.941** | **0.956** | **0.978** | 0.499 |
| Ours | **0.989** | 0.300 | **0.551** | 0.929 | 0.948 | 0.974 | **0.604** |
| **Re-Camera Control** | | | | | | | |
| ReCamMaster-Wan (Bai et al., 2025a) | **0.994** | 0.500 | **0.459** | **0.930** | 0.921 | 0.983 | 0.576 |
| Ours | **0.994** | **0.600** | 0.449 | 0.925 | **0.942** | **0.986** | **0.604** |

### B.2    ADAPTION TO OTHER BASE MODEL

We also adapt our method to another base model Wan-1.3B (Wang et al., 2025a), which has a similar model size and comparable T2V performance to our base model. The experimental results are shown in Table 8. We evaluate our method on three different tasks: ID insertion, stylization, and camera control (ReCam). The results demonstrate that our approach achieves comparable performance across both base models. While the Wan 1.3B model shows slightly lower performance in some metrics, this is likely due to the fact that **we did not further adjust the training parameters specifically for this model**. Nevertheless, these results demonstrate that our method is model-agnostic and successfully unifies these diverse video generation tasks across different base models.

### B.3    BENEFITS OF UNIFICATION

Our approach demonstrates significant advantages over task-specific adapters or task-specific tuning like LoRA or adapters. We analyze these benefits from three key perspectives:

Table 8: Performance comparison on different base models.

| Task | Type | CLIP-I↑ | DINO-I↑ | CLIP-Score↑ |
|---|---|---|---|---|
| ID Insert | Internal 1B | 0.598 | 0.245 | 0.216 |
| ID Insert | Wan 1.3B | 0.528 | 0.210 | 0.193 |
| **Task** | **Type** | **CSD-score↑** | **ArtFID↓** | **CLIP-Score↑** |
| Stylization | Internal 1B | 0.259 | 37.619 | 0.171 |
| Stylization | Wan 1.3B | 0.236 | 35.318 | 0.181 |
| **Task** | **Type** | **RotErr↓** | **TransErr↓** | **CLIP-Score↑** |
| ReCam | Internal 1B | 1.275 | 5.667 | 0.220 |
| ReCam | Wan 1.3B | 1.272 | 5.674 | 0.218 |

**Parameter Efficiency** Our method achieves remarkable parameter efficiency by adding only thousands of trainable parameters to support six distinct tasks. In contrast, using separate LoRA modules would require training and storing multiple task-specific modules, each containing millions of parameters, resulting in substantially higher storage and computational demands.

**Multi-task Synergy** The shared architecture enables positive interactions between tasks through cross-task information sharing. This is particularly evident in addressing domain-specific limitations. For example, in re-camera control tasks, multi-task joint training significantly improves human representation quality compared to single-task training, as shown in Table 9. The multi-task approach achieves better scores in human-anatomy (0.903 vs 0.889), human-identity (0.887 vs 0.844), and human-clothes (0.921 vs 0.891), demonstrating how shared knowledge across tasks helps mitigate domain-specific constraints.

Table 9: Comparison of human-centered metrics between ReCam-only training and multi-task joint training.

| Task | Training Strategy | Human-Anatomy↑ | Human-Identity↑ | Human-Clothes↑ |
|---|---|---|---|---|
| ReCam | ReCam Only Training | 0.889 | 0.844 | 0.891 |
| ReCam | Multi-Task Joint Training | 0.903 | 0.887 | 0.921 |

**Efficient Fine-tuning** The unified architecture facilitates rapid adaptation to new tasks through shared representations. Our experiments demonstrate the efficiency of knowledge transfer, as shown in Table 10. When fine-tuning for propagation tasks based on ID swap knowledge, we achieve superior performance (RefVideo-CLIP: 0.820) with just 0.4k training steps, compared to 2k steps required by propagation-only training (RefVideo-CLIP: 0.792). This 5× reduction in training steps, along with consistent improvements across multiple metrics, validates the effectiveness of our unified framework.

Table 10: Comparison of fine-tuning efficiency and performance metrics between different training strategies.

| Strategy | Training Steps | RefVideo-CLIP↑ | CLIP-score↑ | Smoothness↑ | Dynamic↑ | Aesthetic↑ |
|---|---|---|---|---|---|---|
| propagation only | 1k | 0.765 | 0.205 | 0.948 | 12.956 | 5.028 |
| propagation only | 2k | 0.792 | 0.219 | 0.952 | 13.262 | 4.912 |
| based on ID swap | 0.2k | 0.789 | 0.217 | 0.951 | 13.245 | 5.036 |
| based on ID swap | 0.4k | 0.820 | 0.226 | 0.956 | 13.562 | 5.265 |

These benefits enable not only more efficient model deployment but also better task performance and easier extension to new video editing tasks. Moreover, unlike methods requiring multiple LoRAs, our unified approach allows for seamless task composition without the risk of conflicting behaviors when combining different editing operations.

## B.4 SCALING UP MODEL SIZE

We conduct comprehensive experiments to investigate the impact of model scaling on various video editing tasks. Our analysis compares the performance between 1B and 10B parameter models across different tasks and evaluation metrics, as shown in Table 12.

Table 11: Comparison with different conditioning and tuning methods.

|         | Lightweight | Efficient Finetuning | Task Composition |
|---------|:-----------:|:--------------------:|:----------------:|
| LoRA    | ✓           | ✗                    | ✗                |
| Adapter | ✗           | ✗                    | ✗                |
| Ours    | ✓           | ✓                    | ✓                |

Table 12: Performance comparison between 1B and 10B parameter models across ID tasks.

| Task      | Type | CLIP-I↑ | DINO-I↑ | CLIP-Score↑ |
|-----------|------|---------|---------|-------------|
| ID Insert | 1B   | 0.598   | 0.245   | 0.216       |
| ID Insert | 10B  | 0.632   | 0.287   | 0.246       |
| ID Swap   | 1B   | 0.725   | 0.429   | 0.242       |
| ID Swap   | 10B  | 0.731   | 0.447   | 0.238       |

| Task      | Type | PSNR↑   | RefVideo-CLIP↑ | CLIP-Score↑ |
|-----------|------|---------|----------------|-------------|
| ID Delete | 1B   | 19.171  | 0.900          | 0.217       |
| ID Delete | 10B  | 47.850  | 0.888          | 0.208       |

The results demonstrate significant improvements when scaling up to larger models. For the ID Insert task, scaling to 10B parameters yields substantial enhancements across all metrics, with CLIP-I improving by 5.7% and DINO-I increasing by 17.1%. Similar improvements are observed in ID Swap tasks, while ID Delete shows remarkable enhancement in reconstruction quality with PSNR increasing from 19.171 to 47.850. These results provide strong quantitative evidence that scaling up model size leads to better performance across various video editing tasks.

## B.5 COMPUTATION COST ANALYSIS

We provide a detailed breakdown of the computational requirements for different video editing tasks in our unified framework, as shown in Table 13.

Table 13: Detailed computational requirements for different video editing tasks.

| Task             | Memory Consumption | Inference Time/Iter | Number of Tokens              | Denoising Steps | Video Resolution |
|------------------|--------------------|---------------------|-------------------------------|-----------------|------------------|
| ID Insert        | 24.45G             | 2.45s               | (43, 1008, 1152)=49932288     | 30              | 672x384x77       |
| ID Swap          | 24.45G             | 2.45s               | (43, 1008, 1152)=49932288     | 30              | 672x384x77       |
| ID Delete        | 24.45G             | 2.45s               | (43, 1008, 1152)=49932288     | 30              | 672x384x77       |
| Propagation      | 22.23G             | 2.31s               | (41, 1008, 1152)=47609856     | 30              | 672x384x77       |
| Stylization      | 22.23G             | 2.31s               | (41, 1008, 1152)=47609856     | 30              | 672x384x77       |
| Re-Camera Control| 22.16G             | 3.66s               | (60, 1008, 1152)=69672960     | 30              | 672x384x77       |

## B.6 ANALYSIS OF JOINT TRAINING PERFORMANCE

We investigate the performance degradation observed in joint training and propose potential solutions. Our analysis reveals that the degradation primarily stems from the varying complexity and convergence rates across different tasks. Specifically, re-camera control requires approximately 600k iterations to converge in single-task training, while simpler tasks like ID-swapping and stylization achieve satisfactory results within 80k iterations.

To address this issue, we experiment with an imbalanced sampling strategy that allocates more training resources to the complex re-camera control task (0.7) while reducing the sampling probability for simpler tasks (0.1 each). The results show that while this approach improves re-camera control performance, it leads to degraded performance in simpler tasks, suggesting that achieving optimal performance across tasks of varying complexity remains challenging in joint training scenarios.

## B.7 ANALYSIS OF TASK INDICATION METHODS

We investigate different approaches for indicating tasks in the model input, comparing text-based task indication with our task-aware rope design. Specifically, we explore three settings: (1) pure text

Table 14: Performance comparison between balanced (0.25 each) and imbalanced (ReCam: 0.7, others: 0.1) sampling strategies in joint training.

| Train | ID Swap | | ReCam | | | Style | |
|---|---|---|---|---|---|---|---|
| Strategy | CLIP-I↑ | DINO-I↑ | RotErr↓ | TransErr↓ | CamMC↓ | CSD-Score↑ | ArtFID↓ |
| balanced | 0.713 | 0.421 | 2.287 | 9.694 | 10.377 | 0.298 | 38.953 |
| imbalanced | 0.712 | 0.409 | 2.010 | 7.279 | 8.040 | 0.192 | 39.453 |

indication where we prepend task type to the caption (e.g., "[stylization] A woman is..."), (2) our original design with task-aware rope and condition bias, and (3) a combined approach incorporating both mechanisms.

Table 15: Performance comparison of different task indication methods. Text Indication refers to indicating task type to caption. Ours means using task-aware RoPE and condition bias. Combined refers to indicating both through text and our design.

| Method | Text Indication | Task-aware Rope | Condition Bias | ID Swap | | Stylization | | ReCam | |
|---|---|---|---|---|---|---|---|---|---|
| | | | | DINO-I↑ | Refvideo-CLIP↑ | CSD-Score↑ | ArtFID↓ | RotErr↓ | TransErr↓ |
| A1 (Text Only) | ✓ | | | 0.432 | 0.723 | 0.294 | 42.090 | 1.622 | 6.613 |
| A2 (Ours) | | ✓ | ✓ | 0.429 | 0.776 | 0.259 | 37.619 | 1.275 | 5.667 |
| A3 (Combined) | ✓ | ✓ | ✓ | 0.441 | 0.733 | 0.304 | 40.771 | 1.653 | 6.416 |

While text-based task indication (A1) shows improved condition image similarity metrics (DINO-I and CSD-Score), it compromises video consistency (lower Refvideo-CLIP) and ReCam task performance. The combined approach (A3) also underperforms compared to our original design (A2), suggesting that explicit text indication may interfere with the model's ability to balance instruction understanding and content preservation. This limitation might be addressed through large-scale pre-training.

## C  UNIC BENCHMARK

To comprehensively evaluate performance, we create a unified benchmark of six tasks, each containing 20 to 50 carefully designed evaluation cases.

### C.1  ID INSERT

We collect 20 videos from Artgrid (art, 2025) as source video, and we carefully select a suitable ID for each video to insert, also ensuring that the semantic is reasonable. As shown in Fig.S 4, our selected ID includes both clean object without background and complete picture with background.

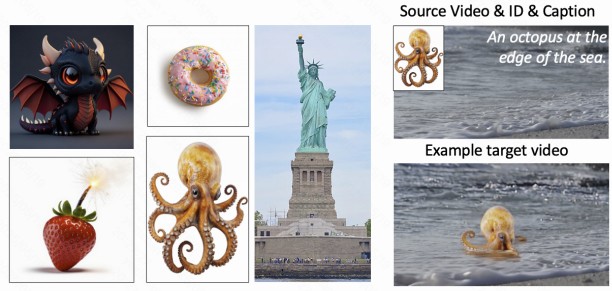

Figure 4: **ID Pool and example of ID insert evaluation cases.**

### C.2  ID SWAP

For this task, we utilize 14 videos from VPBench (Bian et al., 2025b) and 6 videos online as the source videos. With the source videos, the objects to be swapped were segmented using SAM2 (Ravi et al., 2024). Then, we carefully design and choose the object to place in. As shown in Fig.S5, an

appropriate ID was then selected from our ID pool to replace the segmented object, and a caption was generated to describe the final target video.

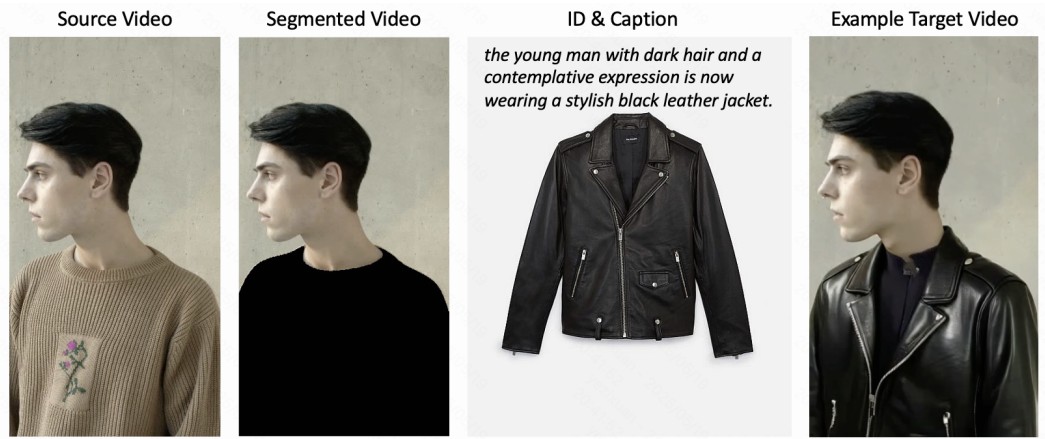

Figure 5: **Example of ID swap evaluation cases.**

### C.3    ID DELETE

For ID delete, we expand the 10 videos in VPBench (Bian et al., 2025b) to 20 by additionally collecting 10 videos. Also, SAM2 (Ravi et al., 2024) is used to segment the object to be deleted. Then we generate caption for the target video. The example is shown in Fig.S6.

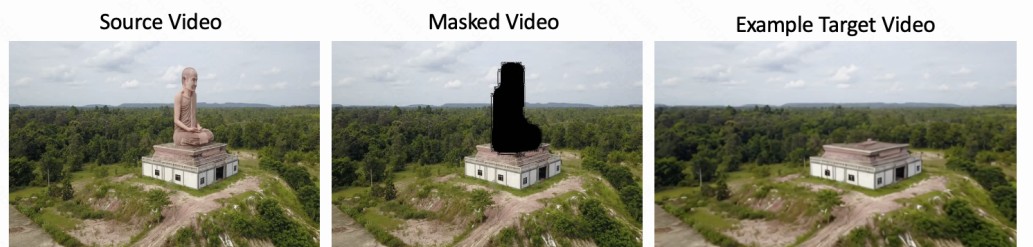

Figure 6: **Example of ID Delete evaluation cases.**

### C.4    STYLIZATION

For the stylization task, we collected 12 representative styles to serve as references. These include diverse artistic expressions such as pixel art, oil painting, Chinese painting, and line art, among others. Examples of these styles are illustrated in Fig.S7. Then we randomly select 50 videos from Artgrid (art, 2025) as the source videos.

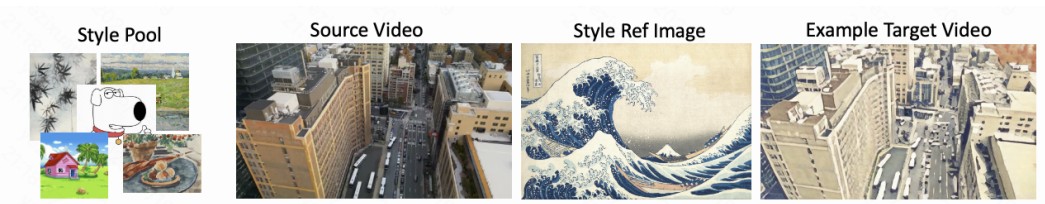

Figure 7: **Example of Stylization evaluation cases.**

## C.5 PROPAGATION

For the propagation task, we expand the 38 example of GenProp (Liu et al., 2024a) to 50 examples by adding 12 stylization propagation test cases. Example is shown in Fig.S8.

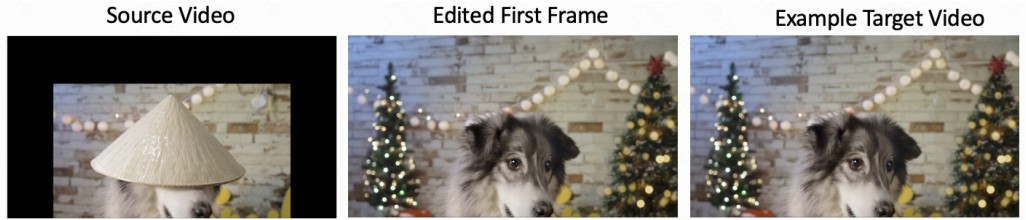

Figure 8: **Example of propagation evaluation cases.**

## C.6 RE-CAMERA CONTROL

To evaluate the re-camera control task, we utilized 10 basic camera trajectories and 50 randomly selected videos from Koala (Wang et al., 2024a). Each of the 10 trajectories was then applied to 5 distinct videos from this set (totaling 50 trajectory-video pairs).

# D TRAINING DATASET CONSTRUCTION

This section details the construction of our datasets for the six tasks.

## D.1 ID-RELATED TASK

To generate training data for ID-related tasks such as deletion, swap, and insertion (as illustrated in Fig. S9), we first use SAM2 (Ravi et al., 2024) to obtain an object's segmentation mask from the source video. This mask is then applied with `cv2.inpaint` to produce an inpainted video. However, this simple inpainting method often introduces visual artifacts in the inpainted regions. To address this, we train a ControlNet conditioned on the original video, which effectively eliminates these artifacts. The resulting artifact-free video serves as the reference video for the insertion task, with the original source video as the target. Additionally, a masked video, created by applying the segmentation mask to the source video, serves as the reference video for both deletion and swap tasks. Using this method, we create 3000 videos for each task.

## D.2 STYLIZATION

When considering how to construct the paired style video dataset, a straightforward idea is to use a video-to-video stylization model to convert real-world videos into stylized ones. However, our experiments revealed that this approach frequently results in temporal inconsistencies, flickering artifacts, and lower visual quality.

We then noted that Text-to-Video (T2V) models are capable of generating stylized videos that exhibit superior quality and maintain higher fidelity to a given reference style image. This observation led us to an alternative strategy: rather than stylizing an existing real video, we first generate a high-quality stylized video using a T2V model. Subsequently, we transform this stylized video into a realistic counterpart using a tile-based video ControlNet. As illustrated in Fig.S10, the results confirm that this is a feasible method. Using this method, we create 10,000 paired videos.

## D.3 PROPAGATION TASK DATASET

To construct a dataset for the propagation task, we leverage existing paired data from our ID-related and Stylization task training sets. Each pair in these datasets contains a source video and a target video.

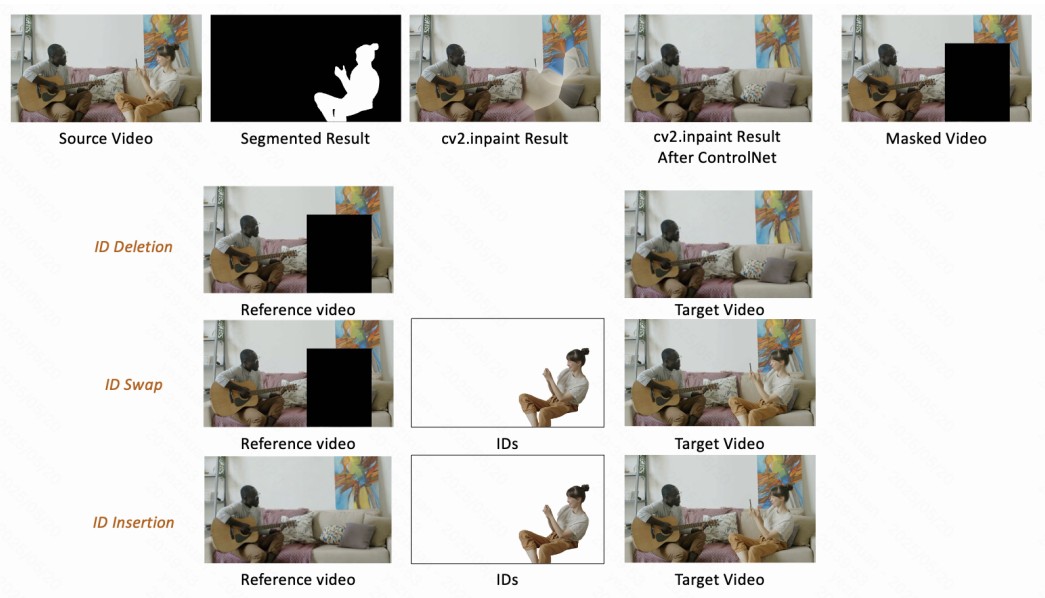

Figure 9: **ID-related task data construction.**

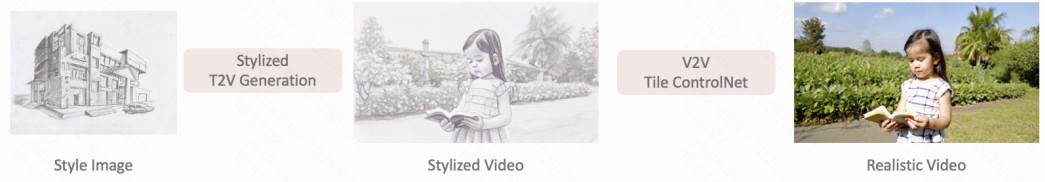

Figure 10: **Stylization pair data construction.**

The propagation task requires input triplets consisting of a source video, a target video, and the first frame of that target video. We can generate these triplets from our existing paired data in two distinct configurations. In one configuration, the original source video serves as the propagation source, the original target video serves as the propagation target, and we use the first frame of this original target video. Alternatively, the roles can be reversed: the original target video can serve as the propagation source, with the original source video becoming the propagation target, and we would then use the first frame of this (now) target video. This strategy effectively allows us to derive a greater volume of propagation training data from our existing resources. Specifically, we combine 9,000 samples from ID-related tasks and 10,000 samples to form a 19,000-sample dataset.

### D.4 RE-CAMERA CONTROL TASK DATASET

For the Re-Camera control task, we employ the Multi-Cam Video dataset from the ReCamMaster (Bai et al., 2025a). This established training dataset provides 136,000 videos.

## E TRAINING SCHEMES

### E.1 MODEL DETAILS FOR SIX TASKS

Figure S11 illustrates the model architecture details for handling the six distinct tasks. The configuration for each task, particularly concerning input encoding and Rotary Positional Encoding (RoPE) indices, is as follows:

- **ID Insertion and Swapping:** For these tasks, the injected ID image(s) are encoded using the same 3D VAE Encoder employed for the reference video. Their RoPE indices commence

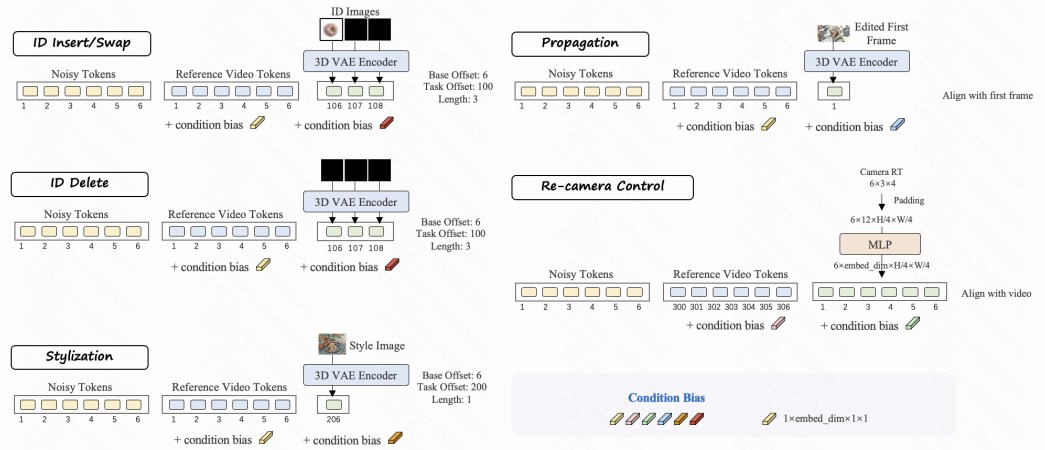

Figure 11: **Detailed Input and RoPE index for the six tasks.**

from a base offset of 6, with an additional task-specific offset of 100. This allocation has a length of 3, accommodating up to three ID images. If fewer than three ID images are provided for a given instance, the embedding slots for the remaining IDs are filled with representations corresponding to black images.

- **ID Deletion:** In the ID deletion task, all input ID image slots are effectively treated as black images (i.e., their embeddings correspond to black images), signaling the removal operation. The RoPE indexing follows the same base offset of 6 and task offset of 100 as ID insertion/swapping.

- **Stylization:** For the stylization task, the style reference image is also embedded using the 3D VAE Encoder. It utilizes the same base RoPE offset of 6, but with a different task-specific offset of 200. Therefore, the RoPE indices for style tokens begin at 206 (i.e., $6 + 200$).

- **Propagation:** The propagation task leverages the direct correspondence with the first frame of the target video. Consequently, the tokens representing this first frame (used as the propagation source/reference) are assigned RoPE index 1.

- **Re-camera Control:** Camera parameters for re-camera control, initially provided as a tensor of size $F \times 3 \times 4$ (where $F$ is the number of frames), undergo a specific tokenization process. First, the last two dimensions ($3 \times 4$) are flattened, resulting in $F \times 12$. These features are then spatially padded to match the VAE token dimensions of $H/4 \times W/4$ to obtain $F \times 12 \times H/4 \times W/4$. Subsequently, an MLP embeds these processed parameters into a tensor of shape $F \times \text{emb\_dim} \times H/4 \times W/4$, aligning them with the dimensionality of tokens encoded by the 3D VAE. Crucially, since this task regards the reference video primarily as soft guidance rather than requiring strict pixel-to-pixel alignment, the RoPE indices for the reference video tokens are shifted by $+300$ from their original positions (e.g., original index $i$ becomes $i + 300$).

### E.2 TRAINING PROGRESS

As shown in Fig.S12, we present two training progression strategies for our method: one starting with tasks deemed "hard" and progressing to "easy" ones, and the converse strategy, from "easy" to "hard". The classification of tasks as "hard" or "easy" is based on our empirical observations during training within this in-context framework. For instance, we found that the re-camera control task typically requires training on approximately 100k data volume to reach satisfactory performance, whereas ID-related tasks (such as ID insertion or swapping) achieve comparable results with only about 20k volume. Consequently, we categorize re-camera control as a "harder" task in this context.

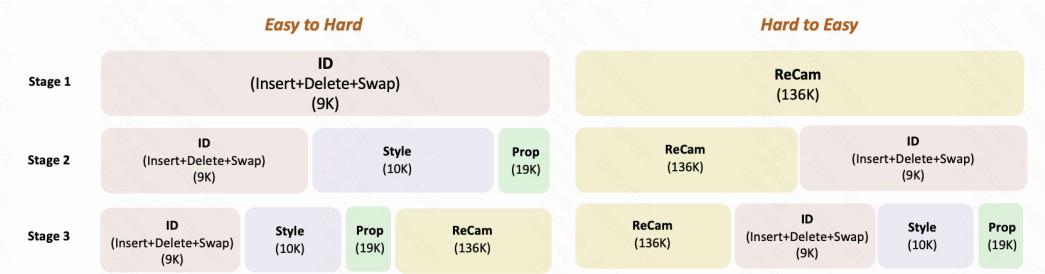

Figure 12: **Training progress with different settings.** We demonstrate two settings of the progress and the data volume of each task.

### E.3 TRAINING DETAILS

UNIC is trained on multiple datasets to support multi-task video editing. All finetuning experiments start from an internal pre-trained model with 1B parameters and 28 sequential standard Diffusion Transformer (DiT) blocks. Each block contains 2D self-attention, 3D self-attention, cross attention and FFN layers.The model is finetuned for 16k iterations on 32 H800 GPUs with a batch size of 64. We only finetune the transformer module and the new tokenizers (like the MLP for the camera pose), while freezing the 3D VAE, Text T5 Tokenizer.

## F STATEMENT FOR LARGE LANGUAGE MODELS

We utilize large language models (LLMs) in this paper only for the purpose of grammar correction and text refinement. The LLMs are not employed for generating original content or contributing to the conceptual development of the ideas presented.

