# OpenReview forum: "Unified In-Context Video Editing"
_ICLR.cc/2026/Conference — ICLR 2026 Poster_

### Official Review · Reviewer_4wrd · 2025-10-29

**Soundness:** 3
**Presentation:** 3
**Contribution:** 3
**Rating:** 6
**Confidence:** 4

**Summary:**

Whereas prior video editing methods often require inefficient video inversion or the design and training of complex control modules, this paper proposes an in-context approach based on a recently introduced DiT-based video generation model. The method concatenates multiple conditional signals and performs full attention over them, enabling a single unified model to handle diverse conditional editing tasks. To further support such a unified model, the authors introduce condition bias and task-aware RoPE, techniques that help distinguish different conditioning signals and improve performance. They also propose a sequential training scheme to make unified training feasible.

**Strengths:**

- **Unified model for diverse video editing tasks:** The paper models various video editing tasks under a single formulation and demonstrates that learning a unified video editing model is feasible. The two new techniques introduced specifically for unified training are effective and constitute an additional strength.
- **Timely extension to video:** As development is progressing from image models toward video models, the proposed work arrives at an opportune time.
- **Clear and intuitive writing:** The paper is easy to follow, and the motivations behind the methods are clear and intuitive.

**Weaknesses:**

- **Slightly limited novelty:** Although the need to distinguish task-specific conditional signals in unified training is well-motivated and the proposed techniques appear appropriate, the overall design feels like a natural extension from a single signal to multiple signals. Moreover, the proposed methods do not seem tightly tailored to video editing, which makes the contribution feel somewhat less novel.

- **Lack of detailed explanation for method and experiments:**
    The sequential training scheme appears crucial to performance based on the presented results, yet it is not discussed in the method section. In addition, the paper does not specify which TTV base model is used during fine-tuning.
    Furthermore, in Table 6, the performance differences between D2–D4 and the baseline (D1) appear marginal depending on the metric, making it difficult to appreciate the improvements. Additional qualitative comparisons would help clarify the effectiveness of the two proposed components.

**Questions:**

- In L473–474, what does “temporal modeling” refer to? Task-aware RoPE seems to improve reference alignment (as Table 6 suggests), whereas “temporal modeling” sounds more like a measure of video quality, making its use here somewhat unclear.
- In L102, what is meant by “index collisions”?

---

> ### Author Response · Authors · 2025-11-22
> **Response to Reviewer 4wrd**
>
> **Dear reviewer 4wrd**,
>
> We thank the reviewer for the thoughtful and valuable feedback. We would like to take this opportunity to clarify the following points and update them accordingly in our revised manuscript:
>
>
> **[Novelty and Designs Tailored for Video Editing]**
>
> We agree that distinguishing task-specific condition signals appears natural in the extension from single to multiple signals, but we believe our design is thoughtful and non-trivial. The core challenge lies in modality disambiguation - where the same modality (e.g., image) serves different semantic functions across tasks (ID reference vs. style reference). We carefully consider: (1) where to inject the bias in the architecture, (2) how to parameterize task-specific biases, and (3) how to efficiently inject the bias. Additionally, our task-aware RoPE is specifically tailored for video editing challenges: it handles frame-alignment across arbitrary video lengths with different tasks, which is important to any-frame-length video editing. Therefore, we believe our method provides a targeted solution for video editing challenges.
>
> **[Discussion of Sequential Training]**
>
> We agree about the importance of sequential training. We have added the introduction of training scheme into method section explaining our progressive training strategy.
>
>
> **[Details of Base Model]**
>
> We are sorry for the missing details. In our experiments, we used a DiT-based video diffusion model with flow matching training. The internal model consists of 28 DiT blocks, containing 2D self-attention, 3D self-attention, cross attention and FFN layers. We also tested our approach on Wan 1.3B and the results are reported in the Appendix B.1, which demonstrates the framework is designed to be model-agnostic and can be applied to other DiT-based video diffusion models. The qualitative results are also updated on anonymous website.
>
> **[Qualitative Comparison about model designs]**
>
> We understand the concern and we have supplemented the qualitative comparison on the anonymous website. Case 1 demonstrates that the condition bias can effectively help distinguish which task the condition refers to. Since our style data contains Van Gogh stylization, without the condition bias, when the model receives the Van Gogh image, it will apply stylization. Instead, with condition bias, the model correctly interprets the image as an identity reference and preserves the subject's appearance without applying unwanted stylistic effects. Additionally, without task-aware RoPE, achieving proper frame alignment during inference becomes challenging because we train with variable-length sequences. The task-aware RoPE mechanism helps address this issue by providing explicit guidance on how to align with the reference video.
>
>
> **[Clarifications]**
>
> - **L473-474 "temporal modeling"**: This means modeling the temporal alignment with the reference video. For the reference video, reusing the RoPE index of the noisy latent can lead to better alignment between the final result and the reference video. Whether the target video should strictly follow a frame-by-frame alignment with the reference video is decided by the task type (e.g., loose alignment in re-camera control and strict alignment in stylization).
>
> - **L102 "index collisions"**: We train with variable-length sequences. Therefore, compared with fixed-length training, the model has to distinguish the start and end of each condition (i.e., reference video, camera pose, noisy latent). This is straightforward in fixed-length training where each condition occupies predetermined, non-overlapping positional ranges. However, it becomes challenging in variable-length training, since the positional indices will collide across different conditions when their lengths vary, making it difficult for the model to correctly identify which condition each position belongs to and where the boundaries between conditions are located.
>
> We appreciate the reviewer for giving thought advice, if you have further advices or concerns, feel free to let us know, we will try our best to solve them.

---

> > ### Comment · Reviewer_4wrd · 2025-11-25
> > **Review of Submission11891 (2)**
> >
> > Thank you for your detailed response and the effort put into the revision. I appreciate the clarifications regarding the base model details and the inclusion of the sequential training scheme in the method section.
> >
> > I have carefully read your response and find your argument on the necessity of modality disambiguation to be reasonable. I have an additional follow-up question to further clarify the technical contributions:
> >
> > - **Design Choices for Condition Bias**: While I understand the rationale behind the Condition Bias, I am curious if you have explored or have insights on alternative design choices. Did you consider other parameterization of condition bias vector? I would appreciate it if you could share any ablation results or empirical insights comparing your learnable embedding approach to other potential designs.
> > - **Clarification on Task-aware RoPE**: Regarding the Task-aware RoPE, you mentioned that it handles frame-alignment across arbitrary video lengths. However, handling arbitrary sequence lengths is generally considered an inherent advantage of RoPE itself, rather than a specific benefit of the "task-aware" modification. In the context of video editing, is the primary advantage of your "task-aware" design specifically the index reuse strategy (assigning the same indices to the reference and noisy video) to force temporal alignment, rather than just handling variable lengths? I would like to clarify the distinct advantage of your approach compared to using standard sequential RoPE, specifically for video domain.
> >
> > Thank you again for your engagement.

---

> > > ### Author Response · Authors · 2025-11-25
> > > **Response to Reviewer 4wrd (2)**
> > >
> > > **Dear reviewer 4wrd**,
> > >
> > > Thank you for your thoughtful follow-up questions. We appreciate your engagement and are happy to provide further clarifications.
> > >
> > > **Design Choices for Condition Bias:**
> > >
> > > We did explore alternative parameterization approaches for the condition bias. Specifically, after the tokenizer, there is a simple MLP layer before entering DiT blocks. We experimented with using distinct MLP layers for different task tokens (e.g., distinct MLP layer for style token and ID token).
> > >
> > > However, we find that for image tokens, reusing the noisy latent's MLP layer weights performs significantly better than using separate MLPs for each task type. This is likely because they operate in the same feature space (image/video), and change the MLP layer will break this property. Here are our ablation results:
> > >
> > > | Task | Type | CLIP-I↑ | DINO-I↑ | CLIP-Score↑ |
> > > |--------|------|---------|---------|-------------|
> > > | ID Insert | condition bias| 0.598 | 0.245 | 0.216 |
> > > | ID Insert | distinct MLP| 0.521 | 0.198 | 0.182 |
> > > | **Task** | **Type** | **CSD-Score↑** | **ArtFID↓** | **CLIP-Score↑** |
> > > | Stylization | condition bias | 0.259 | 37.619 | 0.171 |
> > > | Stylization | distinct MLP| 0.201 | 48.325 | 0.143 |
> > >
> > > Therefore, we choose to add a learnable embedding to the features instead of using linear projections. This approach is simpler and reduces additional parameters while ensuring the performance.
> > >
> > >
> > >
> > > **Clarification on Task-aware RoPE:**
> > >
> > > We agree with the reviewer that handling arbitrary sequence lengths is generally an inherent advantage of RoPE itself. However, there are important distinctions in the video editing context.
> > >
> > > - 1）While sequential RoPE can indeed handle arbitrary lengths in text-to-video (T2V) generation, it faces significant challenges in video-to-video (V2V) editing scenarios. As we mentioned regarding index collision, since we have multiple conditions, the start and end boundaries of each condition token are not clear to the model when using standard sequential RoPE. This ambiguity makes it difficult for pure sequential RoPE to handle the complex multi-modal input structure effectively.
> > >
> > > - 2）Regarding the second concern about our "task-aware" design - the advantage is not only the index reuse strategy, but also assigning indices based on task type. For example, we implement loose alignment for re-camera control and strict alignment for stylization. This task-aware temporal alignment ensures that the model understands the appropriate correspondence requirements for different editing tasks. Also, we think temporal alignment is not a "rather than" relationship with handling variable lengths. The temporal alignment enforcement enables effective handling of variable lengths in the video editing context.
> > >
> > > Therefore, our task-aware RoPE is necessary in both **establishing proper temporal correspondence** between reference and target videos and **enabling robust processing of videos with arbitrary lengths** in multi-modal editing scenarios. These two functionalities make it particularly well-suited for complex video editing tasks where both temporal alignment and length flexibility are crucial.
> > >
> > > We really thank the reviewer for carefully reading our rebuttal and providing thoughtful feedback. We feel very grateful to have this discussion with the reviewer. If you have further advice or concerns, please feel free to let us know, and we will try our best to address them.

---

### Official Review · Reviewer_CmCf · 2025-10-30

**Soundness:** 3
**Presentation:** 3
**Contribution:** 3
**Rating:** 6
**Confidence:** 4

**Summary:**

The paper proposes UNIC, a simple, adapter-free framework that unifies diverse video-editing tasks, such as ID insert/delete/swap, stylization, first-frame propagation, and re-camera control within one model by concatenating three token types (noisy video latent, reference-video tokens, and multi-modal condition tokens) into a single sequence processed by a DiT with full attention. UNIC features two key designs: Condition Bias (task-type embeddings) and Task-aware RoPE (per-task positional indexing), which mitigate task ambiguity and alignment conflicts. On a six-task benchmark, UNIC reports competitive or superior results to task specialists, shows emergent task composition, and includes ablations and efficiency analyses (e.g., step-cache).

**Strengths:**

1. UNIC reformulates many video editing tasks as tokenized conditions concatenated with noisy and reference tokens.
2. UNIC supports six representative tasks and demonstrates emergent compositions (e.g. stylization + re-camera).

**Weaknesses:**

1. I think the paper could benefit from reorganizing its structure for improved presentation. One of the most important parts of training video editing models is in its data construction pipeline, since paired before-and-after editing data is often difficult to obtain. Right now the main text barely has any description of training data, and all of descriptions are in the appendix.
2. Following weakness 1, there are no statistics about the quantity of the training data in the main text. In Appendix D, there are statistics for some tasks, but not all tasks.
3. While it is clear that both condition bias and task-aware RoPE benefit model performance, it seems that combining these two designs leads to weaker performance on the style transfer task, and the results of ArtFID and CFSD are even worse than not having the two designs at all. Can the author provide more intuition on this issue?

**Questions:**

1. In appendix E.3, "All finetuning experiments start from a pre-trained model with 1B parameters and 28 sequential standard Diffusion Transformer (DiT) blocks." What is this base pretrained model?

---

> ### Author Response · Authors · 2025-11-22
> **Response to Reviewer CmCf**
>
> **Dear reviewer CmCf**
>
> We thank the reviewer for the thoughtful feedback, and the recognization on the capability of our method. We address each of your concerns below:
>
> **[Data Construction Pipeline and Statistics]**
>
> We completely agree with the importance of data construction in video editing. However, due to space limitations in the main script, we can only put the detailed description in the Appendix to enable a complete demonstration. We have completed all details of the training dataset in the Appendix, including detailed statistics for all tasks that were previously missing.
>
> **[Performance Analysis on Style Task]**
> Different from other tasks, style transfer requires a balance between preserving content and applying new style. The last two metrics (ArtFID and CFSD) largely evaluate content preservation, but they are not entirely accurate for assessing content preservation in style transfer contexts. To be specific, with these two designs, the model more easily learns style information and injects style into the video. However, this causes larger changes to the content. While these changes are acceptable in certain contexts (such as ignoring some details in line-art stylization), they lead to performance drops in the content preservation metrics.
>
>
> **[Details of Base Model]**
>
> We are sorry for the missing details. In our experiments, we used an internal DiT-based video diffusion model with flow matching training, the details are updated in Appendix E.3. The internal model consists of 28 DiT blocks, containing 2D self-attention, 3D self-attention, cross attention and FFN layers. We also tested our approach on Wan 1.3B and the results are reported in the Appendix B.1, which demonstrates the framework is designed to be model-agnostic and can be applied to other DiT-based video diffusion models. Also, the qualitative results are updated in anonymous website.
>
>
> We really appreciate your advice in refining the paper, If you have further advices or concerns, feel free to let us know, we will try our best to solve them.

---

### Official Review · Reviewer_MK6C · 2025-10-31

**Soundness:** 3
**Presentation:** 2
**Contribution:** 3
**Rating:** 6
**Confidence:** 5

**Summary:**

This paper proposed UNIC, a finetuning framework that enables a video diffusion model to perform a wide range of video editing tasks in an in-context fashion without task-specific modules, adapters, or dedicated inversion pipelines in prior works. A unified video editing benchmark is built and reported UNIC achieves comparable performance with task specialists on the benchmark.

**Strengths:**

S1) The framework is simple that it does not require any architecture changes or inversion overhead. By simply concatenating tokens for any desired editing behavior in finetuning, the framework enables new combinations and hybrid tasks. This highlights the originality.

S2) This paper recognizes and operationalizes the fundamentally different natures of video editing tasks, then abstracts them into a unified, learnable token-based framework. Its treats task diversity as token diversity. The motivation is justified.

S3) Smooth transitions in paper writing.

**Weaknesses:**

W1) Inversion-free approaches like FlowEdit (ICCV 2025) or other direct injection or projection-based methods (where editing is possible without explicit reference-to-noise inversion or heavy architectural mods) are not discussed. The omission is notable, especially as the field moves rapidly toward more general and efficient editing without inversion bottlenecks. Adding the discussion inside would help contexturalize the problem. A more complete landscape would compare: 1) Inversion-based, 2) Adapter-based, 3) Inversion-free, and 4) Unified token-based models.

W2) The paper did not mention which exact video diffusion model was tested in the context. It only mentioned finetuning on a general DiT-based video diffusion model pretrained with flow matching. It remains unclear for us to verify if this can be applied to general cases. Providing the details of the experiment would have enhanced the reproducibility.

**Questions:**

Q1) W2 also raise an interesting question: If such a framework is finetuned on CogVideoX and also Wan 2.2, is it we should expect models with stronger pretrained video diffusion model would perform better? Or the gain is heavily relied on the finetuning? Would it be possible to share some empirical results?

Q2) Why the in-text citations style are inconsistent across the paper?

Style 1 in Introduction: "Current video editing methods primarily follow two strategies to inject reference video and control signals. As depicted in Fig. 2, one stream of methods, represented by Video-P2P (Liu et al., 2024b), AnyV2V (Ku et al., 2024), and FLATTEN (Cong et al., 2023), utilizes DDIM inversion for noise initialization to preserve the main structure of the reference video."

Style 2 Related Works: "Similarly, FLATTEN Cong et al. (2023) utilizes optical flow to identify keypoints and injects their features to maintain motion fidelity. AnyV2V Ku et al. (2024) also leverages spatial, temporal, and CNN features gathered during inversion. While these approaches excel at retaining reference video information, they inherently require an additional stage for inversion, thereby increasing the overall inference cost and computational overhead."

The authors need to fix the in-text citations style (preferably style 1).

---

> ### Author Response · Authors · 2025-11-22
> **Response to Reviewer MK6C**
>
> **Dear reviewer MK6C**,
>
> Thank you for recognizing our motivation and simplicity of our method. We really appreciate your detailed comments and address each of your concerns below:
>
> **[W1: Discussion of Inversion-free Approaches]**
>
> We appreciate the reviewer's suggestion on the discussion of inversion-free methods. However, we would like to clarify that works like FlowEdit are currently mainly focused on image editing, with limited exploration in the video domain. Additionally, these inversion-free approaches typically handle only limited editing tasks such as object or attribute editing, and cannot perform reference-guided editing with visual inputs, which differs from the setting we mainly discuss in the paper. Considering these two points, we would like to temporarily maintain the current categorization because it better reflects the landscape of the video domain. But we follow the advice and add a discussion of inversion-free approaches in the related work section with a brief overview of inversion-free methods.
>
> **[W2: Model Details and Reproducibility]**
>
> We are sorry for the missing details. In our experiments, we used a DiT-based video diffusion model with flow matching training. The internal model consists of 28 DiT blocks, containing 2D self-attention, 3D self-attention, cross attention and FFN layers. We also tested our approach on Wan 1.3B and the results are reported in the Appendix B.1, which demonstrates the framework is designed to be model-agnostic and can be applied to other DiT-based video diffusion models.
>
> **[Q1: Performance with Different Base Models]**
>
> We conducted experiments scaling up to an internal 10B model and also tested on Wan 2.1 1.3B, with results shown in Appendix B.3 and B.1. Since Wan 1.3B demonstrates similar performance in text-to-video (T2V) generation to our internal 1B model, it also achieves similar performance for editing tasks. These two experiments demonstrate that the improvements are not only brought by fine-tuning - with a larger-scale base model, the editing quality and ability can be further enhanced.
>
> **[Q2: Citation Style Inconsistency]**
>
> Thank you for catching this formatting issue. We have standardized all citations to Style 1 format (Author et al., Year) throughout the manuscript for consistency, ensuring professional presentation standards.
>
>
> We really thank the reviewers for the constructive feedback, if you have further advices or concerns, feel free to let us know, we will try our best to solve them.

---

> > ### Comment · Reviewer_MK6C · 2025-11-24
> >
> > Thanks for the rebuttal. FlowEdit although is focused on image editing, has been widely used in video editing. There are also related works using similar methods in video editing.
> >
> > Also the Q1 rebuttal said 10B internal model but on revised appendix its 1B? Typo?
> >
> > I will keep the rating.

---

> > > ### Author Response · Authors · 2025-11-25
> > > **Response to Reviewer MK6C (2)**
> > >
> > > **Dear reviewer MK6C**,
> > >
> > > Thank you for carefully reading our rebuttal. We appreciate your feedback and will address further concerns.
> > >
> > > **[1B vs 10B model]** This is not a typo. The base model used throughout our paper is indeed an internal 1B model. However, in Appendix B.3, we additionally evaluate our method on a larger internal 10B model to provide a more comprehensive comparison and demonstrate the scalability of our approach across different model sizes.
> > >
> > > **[Inversion-free Video Editing]** We sincerely apologize for missing important related work in this area. We acknowledge that FlowEdit and similar methods have indeed been widely applied to video editing tasks. Following the reviewer's direction, we have conducted a more detailed literature review and have updated our related work section to better position our contribution. Since these methods mainly focus on only text-guided editing instead of a broader concept with reference-guided video editing, which is the main focus of our method, we only give a concise discussion.
> > >
> > > We really thank the reviewer for maintaining a positive rating with our work and hope these clarifications will help address the concerns. We sincerely appreciate the reviewer for the effort of reminding us of another stream of video editing paradigms to ensure a comprehensive literature review. If you have further advices or concerns, please feel free to let us know, we will try our best to solve them.

---

### Official Review · Reviewer_njmg · 2025-11-03

**Soundness:** 3
**Presentation:** 3
**Contribution:** 3
**Rating:** 6
**Confidence:** 4

**Summary:**

The paper proposes UNIC (UNified In-Context Video Editing)， a framework for unifying diverse video editing tasks (e.g., ID insertion/deletion/swap, stylization, re-camera control, propagation) within a single diffusion transformer model.
Instead of relying on task-specific adapters or DDIM inversion, UNIC concatenates noisy video, reference video, and multi-modal condition tokens into a single token sequence, enabling unified learning through native transformer attention. Experiments on a unified benchmark of six tasks show that UNIC achieves competitive or superior results to baselines like VACE, AnyV2V, and ReCamMaster, while offering emergent task composition and parameter efficiency.

**Strengths:**

1. The idea of in-context unification across multiple video editing tasks via token concatenation is elegant and clear

2. The introduction of Condition Bias and Task-Aware RoPE is well-motivated and clearly implemented.

3. The benchmark spans six distinct tasks, covering both local (ID editing) and global (stylization, propagation) settings. it is comprehensive .
4. The demonstration of task composition is interesting, highlights that the model generalizes beyond discrete training tasks

**Weaknesses:**

1. The paper doesn’t compare against or reference newer state-of-the-art T2V and editing models like Wan series

2. While the in-context formulation is elegant, much of the architecture directly borrows from existing full-attention DiT and OmniGen paradigms. The novelty primarily lies in combining these ideas for video, rather than a fundamentally new mechanism.

3. Missing some important related work discussion like VEGGIE [1], which is also unified video editing framework + abilities in in-context video editing

4. Some metrics (e.g., CLIP, DINO, ArtFID) are weak proxies for perceptual quality. The paper doesn’t include human evaluation or temporal consistency metrics (like VBench perceptual coherence, even though it is for video generation, it still captures the generation quality of a video editing model).

[1] VEGGIE: Instructional Editing and Reasoning Video Concepts with Grounded Generation. ICCV25.

**Questions:**

please see weaknesses section.

---

> ### Author Response · Authors · 2025-11-22
> **Response to Reviewer njmg**
>
> **Dear reviewer njmg**,
>
>
> **[Comparison with State-of-the-Art T2V and Editing Models]**
>
> We understand your concern about comparisons and references with recent models. Indeed, we have compared with the newest editing model VACE in the Wan series, and we apologize for any missing references of newer state-of-the-art T2V models. Since many recent T2V models focus on T2V generation rather than the unified V2V editing framework we propose, making the performance comparison challenging. Therefore, we reference and discuss them in the related work section, including the most recent works: T2V models (Wan2.2, LongCat) and other editing models (UniVideo, VEGGIE mentioned by the reviewer).
>
> **[Architectural Novelty and Technical Contributions]**
>
> We understand the concern about architectural novelty, and we will clarify it here.
>
> - First, we are not aiming to design a fundamentally new mechanism, but rather to find an optimal and simple solution for video editing. The work OmniGen mentioned by the reviewer, along with other works we discussed in related work, mainly focuses on image generation, which is fundamentally different from the video domain.
>
> - Also, video editing presents unique challenges that have never been systematically explored in image editing, such as handling variable-length sequences that require careful frame alignment—challenges that are absent in image editing. Additionally, while existing image-based methods mainly consider image modality as input, our approach considers multiple modalities including image, video, and non-visual conditions like camera poses. Therefore, directly adapting image-based solutions cannot achieve satisfactory results for video editing tasks. Our work specifically addresses these video-centric challenges through efficient architectural designs tailored for the temporal nature of video data and the diversity of condition types.
>
> **[Missing Related Work - VEGGIE]**
>
> We thank the reviewer for pointing out this important related work. We have included the discussion of VEGGIE in our related work section, and here we will also give a discussion. Our approach differs in that we achieve unification through regarding the conditions as context rather than requiring additional LLM understanding components to process the conditions, making our framework more streamlined while still effective. Besides VEGGIE, we also include other related works like UniVideo to ensure a comprehensive discussion.
>
>
>
>
>
> **[Evaluation Metrics and Human Assessment]**
>
> Following the reviewer's advice, we add evaluations on all quality scores on VBench and report results for 3 tasks in the mainscipt **Section 4.2**. We also conduct pairwise human evaluation with 10 evaluators, comparing our method against task-specific counterparts: VACE for ID swap, StyleMaster for stylization, and RecamMaster-Wan for re-camera control. Each evaluator assessed 10 video pairs based on editing quality and overall video quality.
>
> As shown in Table, our unified approach consistently outperforms specialized counterparts across all tasks, achieving preference rates of 60%-75%.
>
> | Task | Comparison | Editing Quality | Video Quality |
> |------|------------|----------------|---------------|
> | ID Swap | Ours vs. VACE | 78.2% vs. 21.8% | 64.5% vs. 35.5% |
> | Stylization | Ours vs. StyleMaster | 62.1% vs. 37.9% | 69.3% vs. 30.7% |
> | Re-camera Control | Ours vs. RecamMaster-Wan | 65.7% vs. 34.3% | 73.8% vs. 26.2% |
>
> *Note: Percentages represent human preference rates in pairwise comparisons.*
>
>
> We really thank the reviewer for the thoughtful advice, if you have further advices or concerns, feel free to let us know, we will try our best to solve them.

---

### Author Response · Authors · 2025-11-22
**Update on Acknowledgment for Comments, Anonymous Website, Paper Revision**

Dear AC and all reviewers,

The author team of UNIC sincerely appreciates your contributions in handling this submission and helping us with refining the quality of this work. **We are particularly encouraged by the positive consensus (Ratings of 6 from all four reviewers) and reviewers recognition of our work's potential**. We are really pleased to see theat the reviewers acknowledge the following aspects of our work:

1. **The simple and effective unified in-context formulation: the elegance, simplicity, and originality of unifying diverse video editing tasks via a single Diffusion Transformer (DiT) and token concatenation** (Reviewer MK6C, Reviewer 4wrd).

2. **Motivated technical designs: the introduction of condition bias and task-aware RoPE was viewed as well-motivated and effective for robust unified learning**(Reviewer njmg, Reviewer 4wrd).

3. **Comprehensive evaluation: acknowledgement of a comprehensive benchmark covering six distinct editing tasks** (Reviewer njmg, Reviewer CmCf).

Regarding reviewers' concerns, we also have made efforts to address them in the relevant sections and reflect them in the Paper Revision and Updated Anonymous Website.

> **1. Updates on the Anonymous Website**: https://unicedit.github.io/

- **Qualitative Ablation Study of Condition Bias and Task-aware RoPE**: we update a qualitative comparison to address the concerns of the effectiveness of the two proposed components. (Reviewer 4wrd)

- **Performance on Wan-1.3B**: we implement our method on other base model, Wan-1.3B, it also successfully unifies different tasks, and we add the performance on the website.

> **2. Paper Revision**

Based on the reviewers' valuable comments, we revised our paper from the following perspectives:

- **Refining related work part**: we add more discussions and references in the related work part, including the  inversion-free image and video editing work mentioned by Reviewer MK6C, and other video editing work like VEGGIE mentioned by reviewer njmg.

- **Adding training strategy introduction in method part**: we introduce the progressive training strategy in method part, as suggested by Reviewer 4wrd.

- **Adding more video quality metrics in Experiment part**: we include comprehensive VBench video quality evaluation results in Table 2.

- **Adding comprehensive training dataset details**: we supplement detailed statistics for the training dataset in the Appendix.

- **Adding details of our base model**: we supplement the base model details in Appendix E.3.

- **Adding quantitative results on other base model**: we supplement the quantitative comparison on other base model details in Appendix B.1.

---

### Meta-Review · Area_Chair_wvA2 · 2026-01-06

**Summary:**

The reviewers generally agreed on the simplicity and elegance of the proposed framework, noting that the in-context tokenization approach is an effective way to unify diverse video editing tasks without specialized adapters. However, they also have some shared concerns:

1. The original submission omitted the specific base model used and details regarding the data construction pipeline and the "sequential training" strategy.

2. Reviewers pointed out that automated metrics like CLIP and ArtFID are insufficient for capturing temporal consistency and perceptual video quality.

3. Missing discussions on newer state-of-the-art models and alternative paradigms like inversion-free editing and reasoning-based editing.

4. Some confusion existed regarding the performance of Condition Bias and Task-aware RoPE on specific tasks like stylization.

**Reviewer Concerns:**

Most of the concerns were addressed:
1. The authors clarified the use of an internal 1B DiT model and provided new results using the public Wan 1.3B and a larger 10B model, demonstrating that the method is model-agnostic and scalable.

2. In response to R1, the authors conducted a human study with 10 evaluators. The results showed a significant preference (60-75%) for UNIC over task-specific specialists like VACE and StyleMaster.

3. The authors integrated discussions of missing related works such as VEGGIE and FlowEdit, successfully contextualizing UNIC within the broader landscape of inversion-free and instructional editing.

4. The authors provided intuitive explanations and qualitative cases on the anonymous website demonstrating how these components prevent "index collisions" and modality ambiguity.

**Reviewer Scores:**

All reviewers will hold a score larger than 6 as the initial concerns have been addressed.

---

### Decision · Program_Chairs · 2026-01-26

Accept (Poster)